# Situated Pedagogy in Danish Daycare—The Politics of Everyday Life

## Maja Røn-Larsen [1] and Anja Hvidtfeldt Stanek [2,*]

1 Department of People and Technology, Roskilde University, 4000 Roskilde, Denmark; mrl@ruc.dk
2 Department of Psychology, University of Southern Denmark, 5230 Odense, Denmark
* Correspondence: ahstanek@health.sdu.dk

**Abstract:** This article analyzes the possibilities and obstacles in pedagogical practices in ECEC (Early Childhood Education and Care) in relation to developing relevant opportunities for participation for all children, by supporting their own engagements in order to expand their action possibilities. Over the last decades, the political agendas in the Nordic as well as other OECD countries have been led by an increasing focus on learning goals and standardized professional procedures, at the expense of a more situated and flexible pedagogy following children's own engagements. When concerns arise about children's well-being, development, and/or learning, this tendency seems to intensify, as descriptions of concerns are often based on assessments of children's individual (dis-)abilities, while investigations of children's own engagements and reasons for actions are seldom conducted. From a theoretical standpoint in critical psychology and social practice theory, we discuss collaborative processes among children and adults in relation to institutional conditions as inherently political, in the sense that the distribution of different access to social resources and opportunities for participation for different children is negotiated through such daily exchanges and therefore also involves questions about democracy. We explore the everyday life practices of children and professionals, analyzing how, through everyday practice, they constantly work on maintaining, reproducing, and transgressing the standardized demands. To understand such processes, we suggest a conceptual focus on the politics of everyday life and situated pedagogy.

**Keywords:** children's perspectives; children's engagements; democratic pedagogy; everyday politics; expanding action possibilities; opportunities for participation; situated pedagogy

## 1. Introduction—About Situated Pedagogy

*"We have this fourteen-month-old boy who really likes the other children. A very happy, enthusiastic little boy who really wants to be with his friends, and who is practicing a lot in 'how do I do this?'*

*For example, if some children are sitting on the mattress or someone is playing in the toy-kitchen, he often comes up from behind and throws himself on their backs. The others might get a bit frightened by this and think it's a bit intense. However, if you just help him so that he comes from the front and give him a food item or whatever they're playing with, rice bags or... Then he joins in the game.*

*We simply move him physically 'whoop' [with hand gestures she indicates a movement of lifting and turning the child]. Just like that, turns him so he addresses them from the front. It's a tiny little movement, but such a tiny little thing can make a huge difference in the interaction, and in the way he is able to play and participate. In this way, he is welcomed in a completely different way. It's a tiny little thing that makes a huge difference."*

(A Pedagogue's example of "a tiny little thing, that makes a huge difference".)

Different theoretical perspectives have pointed to the importance of children's perspectives and agency in relation to developing the pedagogical practice of daycare (Clark 2020;

Sevón et al. 2023; Warming 2005). According to the United Nations Convention on the Rights of the Child (UNCRC), every child has the right to be heard and express their views in matters that concern them (UNICEF UK 1989). Although children's participation is a statutory requirement of the legislation for daycare in Denmark (EVA 2020), educational professionals in daycare do not always succeed in enhancing children's opportunities for participation, due to different institutional conditions in their work (Plum 2018; Georgeson et al. 2016; Røn-Larsen 2012). Informed by critical psychology and social practice theory, our view on children rests on the basic assumption that children, as all other persons, are oriented towards each other and are directed toward gaining influence and involvement in relation to what happens in their lives. Overlooking this "actively acting part" of human psychology risks reducing pedagogy to a matter of working with the objectified child's adaptation to the existing society, rather than with opportunities for the child's subjective participation in and contribution to society's continuous development and change.. Ultimately, such reduced pedagogy can lead to mental illness and disturbances for the children (Holzkamp-Osterkamp 1991; Stanek 2022). In continuation of this, pedagogical efforts to ensure children's rights and well-being must be connected to both children's own engagements and the social communities they are part of, as well as to an understanding of children as political agents. This implies, in continuation of this Special Issue's theme, an implicit aim for democratic institutions of everyone being involved and everyone having a say in relation to what happens.

The empirical example above illustrates how a pedagogue works concretely and situated in order to support young children in following their engagements in ways that expand action possibilities and opportunities for participation among the other children. It is an educational effort that requires a pedagogue's presence, quick reflections, and caring actions, in order to relate the engagements of the different children and avoid a situation where some children might be startled by sudden or abrupt contact attempts from others. In order to do so, it is necessary to understand a child's engagements in achieving contact with other children, but also to explore the other children's ways of participating, as a basis for the situation, in a relevant and flexible way. Based on the analysis of a child's intentions and the context with other children, it becomes possible for the pedagogue to decode and connect the children's engagements in the shared play. Through this educational intervention, an opportunity is created for children to connect meaningfully with other children's activities. In the specific example, turning the child physically so that he enter from the front, providing him with a remedy relevant to the common play, and helping the others to spot him as someone who can contribute with something instead of someone who scares them, requires complex analytical work. Such a pedagogical approach requires continuous exploration and analysis of the child's intentions, the content of the games, the other children's preoccupations, and possible reactions, in order to find relevant ways to develop conditions for the participation of the different children. However, in the example, the pedagogue describes this as a "tiny little thing".

Both our own and others' research points to this tendency to play down certain aspects of the educational practice as increasingly dominant (Munck and Marschall 2020; Røn-Larsen and Stanek 2015, 2016; Stanek et al. 2018). In this article, we draw on empirical data from different practice research projects conducted together with children and staff in various daycare institutions over the last decades, consisting of participant observations, interviews, and practice research workshops. In general, pedagogical staff's stories about situations like these demonstrate, on the one hand, a pedagogical practice featuring a complex exploration of everyday situations and possible options for professional action. On the other hand, this complexity is often belittled and referred to as "tiny little things" that take place next to "the real" educational work, understood as something that can be measured and documented, hung on the wall, put in folders, or otherwise displayed. Through this article, we analyze the possibilities in pedagogical practices in daycare for supporting children's engagements to expand their action possibilities and agency, in a way where children are supported in gaining influence and, from a first-person perspective,

*relevant* opportunities for societal participation (Røn-Larsen and Stanek 2015, 2016). We will present empirical examples that illustrate how even very young children develop their own engagements through their everyday life and are directed towards expanding their possibilities and access together. We will also show how children's initiatives and commitments are easily overlooked in pedagogical practice, which is increasingly guided by expectations of standardized efforts, focusing on specific aspects of prevention, development, and learning. We will show how "tiny little things" are crucial for a democratic pedagogy/educational practice directed at creating relevant possibilities for all children to be supported in their first-person perspective towards *for them* relevant opportunities for engagement and participation in the world—and we will also illustrate how this pedagogy is under severe pressure.

The political agenda in Denmark over the last few decades has increasingly been led by a focus on learning goals, preventive efforts, and standardized professional procedures at the expense of a more situated flexible pedagogy following children's own engagements (Stanek 2013, 2022). This tendency is exacerbated when concerns arise about children's well-being, development, and/or learning, as descriptions of concerns are often based on assessments of children's individual (dis-)abilities, while investigations of children's own engagements and reasons for actions are seldom conducted (Georgeson et al. 2016; Hvidtfeldt and Stanek 2022).

It seems that the increasing trend towards standardized notions of prevention, quality, and outcome can easily get in the way of the necessary local exploration and collaboration with children. This risks creating unequal conditions for children of concern in particular, as the focus is on tracking down and categorizing "children of concern" and initiating standardized training interventions for their specific problems in ways that overshadow collaboration with children about their specific life situations, including what they are engaged in and what is relevant to them (Juhl 2023).

In other words, current developments seem dominated by a focus on "incapabilities" and compensatory interventions, and this is happening against the overall prevention and equality agenda.

As a response to this trend, this article seeks to analyze and conceptualize often overlooked aspects of pedagogical practice related to a flexible exploration of children's own engagements and conditions as well as a concrete adjustment of the pedagogical interventions based on this, which we term "situated pedagogy" (Stanek et al. 2018). From different studies in daycare settings, we explore the everyday life practices of children and professionals, analyzing how, through their shared and conflictual everyday practice, they constantly work on transgressing standardized demands (Røn-Larsen and Stanek 2015, 2016).

In the article, we discuss how such collaborative processes in institutional conditions can be analyzed as everyday politics, in the sense that the distribution of access to societal resources and different opportunities for participation is negotiated through such daily exchanges. When we describe pedagogical work as being everyday political, it is because the different participation opportunities are regulated through the pedagogical efforts and the collaborative processes between children and professionals. Creating relevant participation opportunities requires insight into children's (different) perspectives, an insight that is complicated by demands for standardized pedagogical efforts. Focusing on the everyday political aspects of professional practice provides knowledge about how pedagogues, through their everyday political actions and organizations, have an impact on which children can obtain access and gain influence in the pedagogical practice that forms the central conditions for their conduct of life, knowledge that again relates to the overall problematics of inequality and democracy in education.

## 2. Methodological and Empirical Work—Theory, Materials, and Methods

Over the past 20 years, collectively and in separate research projects, we have followed the everyday life practices of children and pedagogues through changing political frameworks for everyday pedagogical life in daycares (Højholt and Røn-Larsen 2015,

2021; Røn-Larsen 2017, 2019, 2024; Georgeson et al. 2016; Røn-Larsen and Stanek 2015, 2016; Stanek et al. 2018). A common aspect across these projects is that they were conducted as practice research involving both children and professionals as co-researchers (Højholt and Kousholt 2014, 2019; Motzkau and Schraube 2015). Through close collaboration with children and professionals over time, we have identified key tasks and dilemmas in pedagogical work concerning how the work relates to children's everyday life perspectives. This article is based on analyses of empirical work from different research projects, consisting of participant observations and interviews with professionals in daycare institutions in different Danish municipalities.

We investigated the everyday life of children and staff to identify children's possibilities for participation and contributions in daycare settings, the professional work around this, and the relations between the children's and the professionals' different perspectives.

We now present the basic understanding of children's lives, participation, and conditions for participation that form the starting point for our approach to pedagogical work, as linked to children's own engagements as tied up in social practice rather than pre-set goals for development and learning.

Pedagogical work based on critical psychology focuses on the active societal participating subject and the subject's conditions for participation in the world. The original theoretical foundation of critical psychology is developed from points about how evolutionary bodily motor development has made possible what is conceptualized in critical psychology as "the distinctly human" (Holzkamp 2005). The distinctly human is thus the phenomenon that humans, in a fundamentally different way from other animal species, are societal subjects who actively (re)create and (re)produce their common (own, but also other people's) living conditions. The basic point is that what is special about humans as organisms is that humans can create and have created the common societal project that we all live under and that we all participate in maintaining, developing, and changing. The act of actively changing or actively maintaining and the joint (cooperative) influence on the relevant living conditions is thus the general "human" aspect. This "human" aspect is often overlooked by theories that dominate educational efforts whenever concerns for children's development or learning arise (Stanek 2022). The lack of this "human aspect" in both psychology and politics is precisely where the criticism in critical psychology takes its starting point (Holzkamp-Osterkamp 1991; Teo 1998; Tolman 1994).

The youngest children in daycare settings are also humans in the above sense (Chimirri 2014; Juhl 2023; Røn-Larsen and Stanek 2015; Stanek et al. 2018). What is important, in relation to children's psychological development, is whether the child basically and broadly is understood in relation to this distinctly human activity, directed at the transformation of their own and others' living conditions. It is important for children's well-being that the pedagogical practice is concerned with figuring out how best to support children's agency, in a way where the children's disposal over their life conditions is expanded (in the long term) (Højholt and Kousholt 2018). In this perspective, it is crucial that the pedagogical practice is understood as a place where children learn to understand themselves as someone who can or cannot/must or must not change their living conditions. In other words, the educational practice must be understood as an active player in the general development of children's agency and life opportunities, both currently and in the future (Holzkamp 2024a, 2024b).

Hence, developing a relevant pedagogical strategy presupposes an investigation of the different perspectives of children (Aronsson et al. 2018; Højholt and Schraube 2016; Ulvik and Gulbrandsen 2015). To avoid misunderstandings, we wish to clarify that working with children's perspectives is not just a matter of asking children what they want or wish for in a given situation and then adjusting the pedagogy accordingly. Nor is it a matter of looking at children to decode or categorize what "kind" of child and with what "kind of needs". Rather, it is about aligning the pedagogical efforts to the children's specific life situations and to what they are engaged in. Knowledge about this cannot be defined in general or determined in advance beforehand, it needs to be developed concretely through

investigation together with the children involved. Hence, we conceptualize such pedagogy as "situated pedagogy" (Stanek et al. 2018).

Working with children's perspectives in this approach therefore implies observing *with* rather than looking *at* the children, considering their life conditions and the conditions around them that affect what they may do (Højholt and Kousholt 2014; Røn-Larsen and Stanek 2015; Stanek 2014, 2019). Discovering relevance for children and educational initiatives is about understanding what children are concerned with (their common cause), who they are concerned together with (the community), and what conditions they have to participate in and contribute meaningfully to. It is these conditions that pedagogical efforts must connect to, to create conditions for children to expand their learning opportunities. Since children's engagements and life conditions are constantly moving and changing, the pedagogical object and the relevant pedagogical approaches are thus something that cannot be predicted or determined in advance. It is therefore a central professional task to be situated, exploring, and developing in relation to what is relevant to children's participation, which again is always already inextricably linked to the communities of practice (Chaiklin and Lave 1993; Lave 1997, 2011; Lave and Wenger 1991).

The attention to (unequal) opportunities for participation in children's communities has become a central focus of the daycare pedagogy in Denmark. It is written into the Danish Daycare Act and thus mandatory for Danish daycares to work toward children's participation in children's communities. But it is often unclear what participation entails and how it is possible to observe and work pedagogically with participation opportunities in practice (EVA 2020).

The critical psychological understanding of participation is a very different understanding of participation than perspectives on children's "social skills" or "social competencies" as an attribute of the individual child or perspectives on "relational work" that focus on detached relationships and dynamics between individuals. Instead, the concept of participation is linked to the general human condition where people develop who they are and their personal lives through participation in and across social contexts with other people—both other children and adults (Chimirri 2014; Dreier 2008; Juhl 2023; Røn-Larsen and Stanek 2015).

Peers appear to be very important in children's life in daycare—even for the very young (Røn-Larsen and Stanek 2015; Stanek et al. 2018). Children need each other to navigate and participate meaningfully across the complexities of everyday life. But they have different access to the social resources that are so crucial to their opportunities to thrive, develop, and learn what they are expected to in their complex everyday lives.

However, these (unequal) participation opportunities are created in situations in practice between children and other participants in institutional contexts, and it cannot be predicted when inequality might emerge and which children might end up in vulnerable situations (Højholt and Røn-Larsen 2021). Hence, it must be observed and explored over time how different participation opportunities develop for different children. Pedagogy related to supporting children's opportunities for development, learning, and participation, therefore, implies access to knowledge about what is at stake in the children's shared life across the many contexts of institutional daycare settings. It is crucial to develop knowledge about how the conditions for participation look from the children's perspectives, as well as knowledge about the history behind the problems that can be observed right now—"the social development history of problems" (Schwartz 2019).

## 3. Everyday Political Aspects of Daycare Pedagogy

From our theoretical standpoint, the politics of everyday life is conceptualized as an aspect of the subject's behavior in everyday life aimed at collective negotiation and transformation of shared life conditions (Røn-Larsen 2019, 2024). This perspective is rooted in research traditions that emphasize the dialectical relationship between the individual and the social, drawing more specifically on the German-Danish tradition of critical psychology (Dreier 1997, 2008; Holzkamp 2024a, 2024b; Højholt and Schraube 2016;

Osterkamp and Schraube 2013), Jean Lave's (2011, 2019), social practice theory, and Anna Stetsenko's (2008, 2013) transformative activist stand. These approaches share a common interest in human practice as inherently political in the sense that people participate with engagements in the world—and through this participation dialectically transform both themselves and their shared world.

From a social practice perspective, reproduction and change are intertwined (Kontopodis et al. 2011; Stetsenko and Arievitch 2004). Making things work in social institutions is a conflictual process, related to engagements in making relevant contributions to specific problems, but it is always related to conflictual institutional conditions (Højholt and Røn-Larsen 2021; Stetsenko 2008, 2013). The central issue is that "politics" encompasses not just legislation and government but also what people create together. Hence, the politics of everyday life is related to processes of finding relevant possibilities of handling specific problems through collaboration between different people with different perspectives, always in relation to contradictory institutional conditions. This is why it is relevant—in relation to professional work in societal institutions—to focus on the political aspects of the personal conduct of everyday life. It is a particular point that micro- and macro-political perspectives cannot be explored separately but are connected through the dilemma-filled pedagogical work in everyday life. The overall societal and historical discussions about the purpose of daycare institutions are represented in contradictory legislation, which again places conflicting demands on daycare professionals, playing into the concrete situations of everyday life in daycare; these contradictions are handled in relation to solving concrete problems around children related to contradictory institutional conditions.

Through their practice in daycare centers, pedagogues and children participate in the negotiation of how children are positioned differently in their everyday lives, with different access to participation and social resources. That is, through their everyday activities, the pedagogues are always already part of the societal negotiations about what and who the pedagogical practice should include and be intended for.

The work that pedagogues perform can therefore be analyzed as inherently political, rooted in the shared but conflicting ambition of creating meaningful and relevant opportunities for participation for all children.

When we wish to understand children's concrete possibilities for expanding their opportunities of participation and influence in their everyday lives, we must therefore take a closer look at the local, situated, and conflictual collaboration and negotiation processes in the educational setting and between different adults and children. One point is to focus on the production of *both* inequality *and* relevant development and learning opportunities for all as *part of* the pedagogical practice in institutional life. This ambiguity must always be analyzed in relation to the structural institutional conditions for the situated early childhood pedagogy.

## 4. Institutional Conditions for Daycare Pedagogy in Denmark

Nordic countries are characterized by a high rate of infants and toddlers attending daycare—an inclination linked to a high rate of women in the labor market and a well-developed national framework for public out-of-family care. In Denmark, more than 90% of 0–2-year-olds attend some form of publicly arranged daycare or a daycare nursery. Almost all children attend kindergarten (Gulløv 2023). The organization of daycare institutions can vary but is often structured with smaller age groups from 0–2 years and larger groups of children aged 3–5 years. These groups may be in different locations or grouped together in larger age-integrated institutions. Daycare centers will employ both trained pedagogues and pedagogical assistants, with and without formal educational backgrounds.

The described everyday political endeavors of a situated pedagogy are under pressure due to increased demands of documentation as well as requirements for models and manualization (Munck 2020; Plum 2013; Togsverd 2023; Togsverd and Aabro 2021). Such non-situation-specific standards can create difficulties for some children to access participation opportunities (Ringsmose and Svinth 2019; Petersen 2009; Vik 2014), and early

intervention models risk being counterproductive and contribute to the very creation of "children and families of concern" (Nilsen 2017; Munck and Marschall 2020). This tendency is intensified by shifts and transitions where problematizations and categorizations come into play. The official requirement to "detect problems before they occur" can lead to a focus on children's individual competency development in ways that lead to pedagogical staff overlooking the complex sequence of events in everyday life as a contextual framework for children's actions. This can be seen in nurseries, for example, where the focus is on language acquisition, motor development, and emotion regulation at the expense of children's engaged participation in child communities. For the time being, the children's area is characterized by more and more age divisions and readiness processes, which constitute vulnerable situations both for the adults' collaboration and assessments and for the children's personal grip on life (Bendix-Olsen 2019; Stanek 2014, 2019).

## 5. Analysis of Different Educational Situations

> *"This is really the biggest exercise of all when you grow into a nursery. You grow into this community, which of course has some norms, routines, and rules. But they're not rules for the sake of rules, they're rules for the sake of the community. And we adjust [pedagogically] to the community [of children], because there are times when we have children who can handle a lot and times when we have children who can handle less. And their community will reflect that at any given time."*

(Quote from an interview with a pedagogue, Stanek et al. (2018).)

Having a focus on children's communities links to the theoretical starting point of our research. When we explore the daycare with an eye to children's communities, we see the processes through which the institutional space is created and developed by both the participating children and adults. When we understand humans as societal beings who must always be understood in relation to the social practice they take part in, then we also understand humans as active (co)creators of the social practice they are part of. On that basis, we understand the concept of community in a very concrete way, as a reminder that we—old and young—create our common world together, and these are the processes we are curious about in the analysis of everyday life politics in educational work. In the following, we will take a closer look at how very young children take part in communities in the nursery or daycare, and together with the adults, contribute to developing the social context that they share and thereby create the everyday life they have in common. As part of that community, many of the conditions through which they develop also come into being (see also Munck 2020). As a pedagogue said in an interview, "They become 'someone' together. I become 'me' together with my mates".

The social institutions are subject to a number of conditions and terms through, for example, legislation and municipal administration, but during our various projects, it became clear that daycare may also, to a large extent, be understood through the concrete and flexible structures that the participants make with each other in everyday life. The daycare facilities are framed by several institutional requirements and conditions, which the children and adults who populate them on a daily basis continuously negotiate together. The professionals describe how everyday life, rules, and routines must be changed and adjusted in cooperation with the children who are currently enrolled in the daycare. Daycare is an ever-changing practice, because children are different and continually changing. In interviews, the professionals also emphasize how the children adjust in relation to each other, learn from each other, and can participate in different things together and in relation to each other.

The following is an observation from a nursery group where the morning fruit has just been served. The children sit around a large table and eat bread and fruit. Along the way, the adult guides the children in terms of ensure the food is passed on. As we shall see, the children are very interested in each other.

*"Magnus and Albert, who are both just over 1½ years old, used to run around together and are now sitting next to each other at the high table. Albert pushes Magnus with his arm as he laughs and leans over to METTE. Magnus whines and Albert pushes again while looking at Magnus. Magnus whines a little louder, and METTE looks at them. She tells Magnus to say "STOP Albert", with his hand raised to mark Stop—a strategy that is brought into play by the adults many times during this day. At first, Magnus cringes a little, while Albert laughingly pushes him again. It doesn't look violent, perhaps because Albert is simultaneously focusing his attention on balancing a grape and a piece of pineapple in his mouth. Magnus turns his back on Albert and looks towards Louis on the other side. A little later, it seems that he is practicing that stop sign a little after all. He looks at Louis and holds his hand up a little—"Bop" he whispers and looks down again. Louis doesn't take note of it, and it doesn't really seem like Louis has done anything that Magnus might be interested in "stopping". The first children are about to finish their fruit and start squirming on the chairs: "Now I don't think we can ask for more from you"—says MARIE out loud. The children leave the table little by little. Magnus probably doesn't want the rest of his bun, which he pushes around on the wax cloth with his index finger. He looks at Albert, who continues eating happily. "That you, that bun," says Magnus appealingly to Albert, and pushes the bun in his direction. Albert takes the bun with the hand facing away from Magnus. Magnus leans over him and twists a piece of the bun, which he puts in his mouth. Albert puts the rest of the bun between his teeth and holds it like a dog would hold a bone as he gets up in his chair. Magnus looks at Albert and narrows his eyes to make fun while he eats the piece of bun that he took back. A little later he climbs down from the chair and runs over to some children who are running back and forth between the mattress and the door to the wardrobe".*

(Excerpt from observation notes, Stanek et al. 2018.)

The example shows us the children's engagement in participating together. It shows us, for example, how children can sometimes tease and push each other to invite them to play. It also shows us through Magnus' expression how difficult it can be to say stop to someone you want to play with. We see how children constantly work to overcome conflicts to expand their continued opportunities for participation, as in Magnus's invitation with the bun to Albert, and Albert's contribution in fooling around with it. The adults also contribute here. When METTE tells Magnus to tell Albert to stop, she tells Albert both that Magnus does not like him pushing and that Albert has to stop pushing. METTE also tells Magnus that it is ok to speak up when something happens that he does not like. METTE thus helps—together with Albert and Magnus—to define the community in the nursery as one where we should not do things to others that they do not like, but also that the community in the nursery is a place where we can talk about boundaries for our likes and dislikes in ways where we can continue to be interested in each other. This is an example of how educational practice can support the development of the community and of how the politics of everyday life take place. In the example, there are adults close to the children who help them find and express the limits and possibilities of ways of being together. Reality does not always look like this, and the community in the daycare is always up for negotiation and under development.

Concerns about children's learning and development often arise in connection with their ways of participating in social contexts. However, educational practice does not always succeed in grasping their ways of participation as the pedagogue in the initial example did. Oftentimes, the teasing and pushing would be understood as more problematic developmental behavior, and "tiny little things" could turn into "big cases", which involve many resources. Often, when concerns escalate, there is an increasing tendency towards overlooking the children's perspectives, engagements, and (lack of) opportunities for participation with other children.

The following excerpts form an example of how the professionals in the daycare center negotiate resources and frameworks for a young child on a day-to-day basis and how the

entire system monitors a child's development from a very early age in an attempt to help them along the way.

The boy whom we call Malte (Hvidtfeldt and Stanek 2022) is described by the pedagogues as "physically stressed"; they see signs of this in the way he breathes and how he lulls himself to sleep at naptime. Therefore, the professionals try to create a setting in order to relieve this physical stress by altering a former storage room to an extra room for Malte and another child who was assessed to have developmental challenges, with permanently assigned support staff. At times, the daycare center pays for this support from their own basic funds, i.e., without allocated extra funds from the municipality; the associated educational psychologist finds it impossible to perform the testing of language and cognitive levels that is necessary in order to gain the additional financial resources for the daycare center.

The head of the daycare center is working with the municipal system to obtain resources and opportunities to support the children with special challenges as well as to gain the families' trust in order for them to collaborate with the personnel in the daycare center. Sometimes, she visits the families at their homes for informal "coffee meetings" when she senses that they might back down on invitations to large formal meetings with Educational Psychologists or other municipal support services. She also allows for a vulnerable family to make use of one of the pedagogical assistants of the daycare as a private babysitter (at the daycare's expense), to allow the parents to go out to dinner or just go for a walk together on a Friday evening. In parallel with the engaged head of the daycare center's intense work in creating conditions for children's participation and development, including resource-intensive efforts from the daycare center, Malte's issues are addressed using a rather individualized approach in the pedagogical practice. In addition, the daycare head's everyday pedagogical effort is restricted by the lack of staff—especially pedagogically trained staff—often limiting the pedagogical attempts to address and connect with Malte's perspectives and his challenges in everyday practice, which the following research observation from a day in a daycare around Halloween illustrates. At this point, Malte has turned 4 years old and has recently been moved from the former storage room to an ordinary kindergarten group. He is no longer necessarily understood as stressed but as a child with language and behavioral disabilities. The pedagogues are still waiting for a psychologist to observe and conduct cognitive tests. Meanwhile, they are insecure about "how to develop 'on' him" (quote from an interview with a pedagogue, (Hvidtfeldt and Stanek 2022)).

> *Malte jumps and fools around a bit by himself with light in his eyes and a smile on his face. He gets involved with an adult in preparing the Halloween decorations. He doesn't play with the other children from the group, and none of the staff tries to support him in doing so. He notices two other boys playing Spiderman and he chooses to cut out a 'Halloween spider' from paper. He gets help with the scissors. Shortly after he sits down quietly and alone at a table and draws. He repeats the word "Spiderman" several times, but none of the adults seem to notice. The other two boys call Spiderman "Spidy". One of them is "Spidy", while the other boy is "spider web" (he is wearing a t-shirt with a spider web illustration). The two boys crawl around and say "Spidy" and "spider", while Malte looks at them with interest. He looks a little puzzled and says—mostly to himself—"Spiderman" and throws hand signs like Spiderman does when he throws his cobweb threads.*

> *Again, none of the adults seem to see or hear what he says and does, and do not support or help him to connect with the other two boys' play. Malte generally doesn't talk much, but he says understandable words and sentences with more than one word. It's no more difficult to understand what he expresses than many of the other children in the group.*

> At lunch:

> *The group eats in the daycare's common room at several smaller tables. To begin with, Malte sits at a table without any adults. He opens his lunch box by himself and starts eating. A little later, an assistant sits down at that table. She talks with the other children*

*at the table, while Malte sits 'by himself' and eats in peace. At one point he drops a meatball on the floor and it rolls under the adult's chair—a chair on wheels, on which the adult sits and rolls a little back and forth. The adults are busy with other children and conversations, and don't notice Malte's meatball on the floor. Malte does not say anything but climbs down from his chair and stands looking at the meatball. He looks like he is wondering if it is relevant or possible to pick up the meatball again. Another adult comes by and sees him standing there on the floor and decides that he must be tired. She does not ask him why he is standing there or what he is looking at, she just takes him by the hand and tells the others that she will be putting him to sleep.*

Throughout this morning's observations, Malte appears to be a child the adults talk "about" but not really talk "with". At one point, two of the assistants exchange "over his head", saying that they think he usually eats more. They talk about the food in his box and about what he chooses to eat first. However, they do not talk to him about his food or about why he did not eat much today.

There is always something at stake for children in their everyday lives that is relevant to take an interest in if you wish to work pedagogically to support their participation, development, learning, and well-being. However, for some children (those with expected "special needs"), this focus on general aspects of "the distinctly human" seems to fade from the pedagogical attention.

For Malte, this means that he is not supported, and sometimes even interrupted, in the engagements he pursues in his everyday life. In this way, the pedagogical efforts contribute further to the troublesome conditions of his relevant participation.

## 6. Results and Conclusions

Through the analyses of this article, we illustrate how "tiny little things" are crucial for a democratic pedagogy/educational practice directed at creating relevant possibilities for all children and how such pedagogy relies on professionals' exploration of the children's perspectives to relate to relevant opportunities for engagement and participation in the world *for the children*. We show how such pedagogy is made possible through the professionals' politics of everyday life in practice, where they arrange and rearrange the practice through constant investigation and flexibility concerning what is at stake for the different participants in the children's communities "at any given time". We also illustrate how this everyday life politics varies, closely connected to the pedagogues' working conditions. The situated pedagogy concerning the important "tiny little things" is under severe pressure, especially for children, for whom there are concerns. We illustrate that when concerns about children arise, there is a tendency for pedagogical efforts and available resources to focus on individual assessments and interventions, rather than supporting children's own engagements and supporting children in what concerns them.

It is a key point that this shift in focus undermines the perspectives of these "children of concern"—that in the flexible, focused professionalism, there are several unused opportunities that these children could benefit from. At the same time, we can see that when these children's perspectives are overlooked in pedagogical practice, it potentially leads to a democratic problem, as they are deprived of the development of agency and influence in their lives. Hence, the efforts and interventions, contrary to their intended purpose, potentially risk reinforcing the unequal opportunities they are meant to prevent and transcend.

**Author Contributions:** Conceptualization, M.R.-L. and A.H.S.; methodology, M.R.-L. and A.H.S.; validation, M.R.-L. and A.H.S.; formal analysis, M.R.-L. and A.H.S.; investigation, M.R.-L. and A.H.S.; resources, M.R.-L. and A.H.S.; data curation, M.R.-L. and A.H.S.; writing—original draft preparation, M.R.-L. and A.H.S.; writing—review and editing, M.R.-L. and A.H.S.; visualization, M.R.-L. and A.H.S.; supervision, M.R.-L. and A.H.S.; project administration, M.R.-L. and A.H.S. All authors have read and agreed to the published version of the manuscript.

**Funding:** This article received no external funding.

**Institutional Review Board Statement:** Ethical approval was waived for this study. In accordance with a statement from Kirsten Kyvik, the chair of the Research Ethics Committee (REC) from the University of Southern Denmark (SDU), the study obeys the Danish law, since no ethical approval is demanded for this type of research project. Accordingly, no Danish authority exists, where approval could be obtained. However, the research projects that form the basis of the article's analyses have been thoroughly reviewed and discussed in the research group: Subject, Technology and social Practice (STP)—Roskilde University: https://forskning.ruc.dk/en/organisations/subjekt-teknologi-og-social-praksis (accessed on 8 February 2024) and in the cross-institutional research group: Practice Research on Development (PIU) https://typo3.ruc.dk/nc/en/departments/department-of-people-and-technology-dpt/research/projekter/practice-research-on-development/ (accessed on 8 February 2024).

**Informed Consent Statement:** Informed consent was obtained from all subjects involved in the studies.

**Data Availability Statement:** The data presented in this study are available on request from the corresponding author due to privacy issues.

**Conflicts of Interest:** The authors declare no conflict of interest.

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
