# Peer review of "Situated Pedagogy in Danish Daycare—The Politics of Everyday Life"

_socsci, doi:10.3390/socsci13020118_

Round 1

Reviewer 1 Report

Comments and Suggestions for Authors

Congratulations on an interesting and engaging paper. I enjoyed reading it and look forward to seeing it published. In order for the paper to be well understood by an international audience some context needs to be provided for the Danish education system, particularly more detail provided about the qualifications and working conditions of pedagogues in daycare in Denmark. What age group do they work in, are there separate groups for different age bands, different qualifications for different age bands etc? I recognise word count may be an issue so a succinct table could work well also to reduce words but clearly explain the current context. 

More detail is needed about the methodology and data collection of the study, it appears that the data is from a range of studies from the authors and this needs to be more clearly articulated early on in the paper. 

Otherwise, I have attached a PDF of the paper with my comments of suggestions to change and adjust. 

Comments on the Quality of English Language

There are some minor changes to make to English in the document which I have included in the PDF. 

Author Response

First of all, we would like to thank you for your overwhelmingly positive and thorough review of our article - We are very pleased that we have managed to convey the article's messages in a way that has enabled readers to understand our points.

In the following, we will outline how we have further developed the article based on your highly relevant comments.

You have suggested linguistic corrections, and have taken the trouble to comment throughout the manuscript. Thank you for these detailed requirements - we have followed all the directions.

You asks for elaboration on the context of the Danish education system. As the you also points out, we are short of space to expand, and think that a table might be counterproductive, as practices vary a lot from municipality to municipality - but we have tried to write a little more context into the section: “Institutional conditions for daycare pedagogy in Denmark” on p. 6.

You also asks for methodology and data to be expanded. Because our analyses unfold across several different projects' material, it would not be appropriate from our perspective to present all the details, as this could easily lead to the points drowning in details - instead, we have unfolded the common features of the projects - both in the introduction to the article and in the methodology section p. 3ff.

Finally, we have checked, unfolded and reduced (especially the self-)references, where it has been possible without disturbing the basis of the article.

Reviewer 2 Report

Comments and Suggestions for Authors

Thank you for giving me the opportunity to read your article.  I really enjoyed it. I have just a few comments and questions for you to consider: First, as far as language is concerned, take a close look at errors on lines 37,54,56 and 62.

Also I ask you to consider the child as "a/the political child" and what that might do with notions of - or conceptualization of the child and if this might influence on your choice of vocabulary.  I think of eg. the concept of participation. Without considering the child as political participation might turn out to be something the adults give to the child and not an activity based on agency.

On page 9, lines 379-383 you state that "Often when the understanding of children's participation....". Is this generally speaking or based on your example?  Please expand a little. Further, re my comment above, you could expand on this (participation) considerably if you discuss the political child and the agentic forces of a child as subject in collectivity.

Comments on the Quality of English Language

See comment above

Author Response

First of all, we would like to thank you for your overwhelmingly positive and thorough review of our article - We are very pleased that we have managed to convey the article's messages in a way that has enabled readers to understand our points.

You have suggested linguistic corrections and pointed out specific points - we have followed all the directions.

You posed a very relevant challenge, with a request to consider the child as "a/the political child" in order to sharpen especially our conceptualization of “participation”. Initially, that made us want to write a completely new and different article, but that is probably within the scope of the time frame for this special issue. Instead, we have worked through our article in relation to detect and edit places where it is, just as the reviewer points out, possible to misunderstand the concept of participation as an instrumental task for educators in relation to creating and offering opportunities for participation available to children.

Finally, we have checked, unfolded and reduced (especially the self-)references, where it has been possible without disturbing the basis of the article.